# Resumming Post-Minkowskian and Post-Newtonian gravitational waveform expansions

Andrea Cipriani,[1, *] Giorgio Di Russo,[2, †] Francesco Fucito,[1, ‡] Jose Francisco Morales,[1, §] Hasmik Poghosyan,[3, ¶] and Rubik Poghossian[3, **]

[1]*Sezione INFN "Roma Tor Vergata" and Dipartimento di Fisica,*
*Università di Roma "Tor Vergata", Via della Ricerca Scientifica 1, 00133, Roma, Italy*
[2]*School of Fundamental Physics and Mathematical Sciences,*
*Hangzhou Institute for Advanced Study, UCAS, Hangzhou 310024, China*
[3]*Yerevan Physics Institute, Alikhanian Br. 2, 0036 Yerevan, Armenia*

The gravitational waveform emitted by a particle moving in a Schwarzschild geometry can be computed, in perturbation theory, as a double series expansion in the limit where distances are large with respect to the size of the horizon, $z = 2M/r \ll 1$ (Post-Minkowskian approximation), but small with respect to the gravitational wavelength, $y = 2i\omega r \ll 1$ (Post-Newtonian approximation). While in the case of bounded systems the two expansions are linked, for scattering processes they describe complementary regions of spacetime. In this paper, we derive all order formulae for those two expansions of the waveform, where the entire dependence on the $y$ or $z$ variables is resummed under the assumption $\omega M \ll 1$. The results are based on a novel hypergeometric representation of the confluent Heun functions and provide a new derivation of the recently discovered Heun connection formulae. Similar results are found for the non-confluent Heun case. Finally, in the case of circular orbits we compute our formulae at order 30PN and show excellent agreement against solutions obtained via numerical integrations and the MST method.

## CONTENTS

* andrea.cipriani@roma2.infn.it
† gdr794@ucas.ac.cn
‡ fucito@roma2.infn.it
§ morales@roma2.infn.it
¶ h.poghosyan@yerphi.am
** poghos@yerphi.am

## I. INTRODUCTION

Recent advances in the sensitivity of experimental apparatus for detecting gravitational waves (GWs) necessitate increasingly precise theoretical computations. The next generation of space-based experiments, such as the Laser Interferometer Space Antenna (LISA), promises to revolutionize the GW detection by significantly enhancing sensitivity and exploring entirely new wavelength ranges compared to ground-based interferometers [1]. LISA will detect long-wavelength GWs originating from sources like supermassive black hole mergers, extreme mass-ratio inspirals (EMRIs), galactic binaries, and the Stochastic Gravitational Wave Background—sources largely inaccessible to ground-based observatories.

The gravitational signals from these sources are diverse. For instance, GWs from supermassive black hole mergers will produce strong signals with large signal-to-noise ratios, while those from EMRIs, though weaker, will carry exceptionally detailed information about the gravitational sources. Indeed, EMRIs waveforms, spanning thousands of cycles during their inspiral phase, provide a unique opportunity to probe fundamental questions about the nature of gravitational sources, including their compactness, the presence or absence of horizons, light rings, echoes [2], and the validity of Einstein's theory of gravity and its possible extensions [3, 4].

Two main analytical tools have been developed for studying the problem we just described: the multipole

expansion [5–7] and perturbation theory [8–19]. The latter, which applies to EMRI systems, is the focus of this paper. In this framework, GWs are computed by solving the Einstein equations to linear order in the perturbations induced by the motion of a lighter body orbiting around a very massive black hole (BH). The radial and angular propagations of the waves are governed by ordinary differential equations of the confluent Heun type. Analytic solutions of these equations are not known, but it is possible to write them as an expansion in the Newton constant (Post-Minkowskian, or PM, approximation) or in the orbital velocity (Post-Newtonian, or PN, approximation).

In the case of circular orbits, the PM and PN approximations are linked and characterised by a single expansion parameter $v$ (the tangential velocity). The current state-of-the-art for the emitted energy in perturbation theory is the 22nd PN (order $v^{44}$) computation in [20]. The expansion is known to be converging slowly [9, 21] due to the presence of logarithmic terms in the velocity, see also [22–25]. In [26–34] novel techniques, capitalizing on recent advances in quantum field theories, have been exploited to derive an exact formula for the Heun connection matrix and alternative formulations of the waveforms were proposed in terms of the instanton partition function of a $SU(2)^2$ quiver gauge theory. Following this path the infinite tower of logarithmic terms (tails and tails of tails) have been resummed into exponentials.

More recently, multipole [35–48] and perturbation theory [49] methods have been applied to the study of scattering processes (open trajectories) in gravity [50, 51], a field where we can largely benefit from the impressive developments and refinements of scattering amplitude techniques in the last few years (see [52] for a review). These systems provide a richer set up, where the couplings of the PM and PN expansions are no longer linked to each other. For example, in the scattering amplitude language, the PM expansion corresponds to the expansion in loops. At a given order in the coupling, the amplitude is exact in the velocity, i.e. it resums the infinite PN series. To reproduce this result in perturbation theory, one has to resum the double expansion in one of the two variables.

For simplicity we focus on the geodesic motion in a Schwarzschild geometry and parametrize the mass, wave-frequency and radial distance by the two dimensionless combinations $z = 2M/r$ and $y = 2i\omega r$, assuming $x = 4i\omega M$ is small. We will show that, at order $x^k$, the dependence of the waveform on the variable $y$ or $z$ can be resummed and written in terms of a single hypergeometric function and its derivative, with coefficients given by two polynomials of degree $k$, that can be determined recursively. The two limits $x \to 0$ and $y$ or $z$ finite lead to two alternative hypergeometric representations of the confluent Heun function, describing the PM and PN approximations respectively. The parameters of the hypergeometric function and the coefficients of the polynomials are all expressed in terms of a characteristic function

$a(\ell)$ that, in the MST [10–19] formulation of the confluent Heun solution, is the "renormalized angular momentum", while in the description of this paper [26–34, 49], it is the Seiberg-Witten (SW) period of the $\mathcal{N} = 2$ supersymmetric gauge theory. In this framework the gauge couplings of the quiver gauge theory are connected to the PM and PN expansion parameters as $q_1 = z$ and $q_2 = y$. The same techniques apply to D-branes, topological stars and other gravitational backgrounds governed by an equation of the Heun type. See [53, 54] for a recent application to the case of topological stars.

To keep the material self-contained, in this paper we present a derivation of the Heun solutions from first principles with no reference to supersymmetric gauge theory or localization, which sit in the background as inspiration to some of our ansätze, that nonetheless could be also accepted as such. These results provide a first principle proof of the Heun connection formula derived in [26–29] from the crossing symmetry of the conformal blocks of a conformal field theory (CFT) . Our results provide an algorithm which is very efficient when implemented in a Mathematica code and that will allow us to compute the gravitational waveform in the case of circular orbits up to the 30th PN order. This latter result will be tested against a numerical solution and the 22nd PN order computation available in literature [20]. Numerical studies of unbounded systems are postponed to future investigations.

This is the plan of the paper: in Section II we derive the PM and PN expansion of the confluent Heun function. In Section III we apply the results of Section II to the study of the gravitational waveforms emitted by a particle moving in a Schwarzschild background. In Section IV we test our results against those obtained via the MST method and numerical integration. In Section V we extend the results to include the case of the non-confluent Heun function. Section VI contains our conclusions.

## II. THE CONFLUENT HEUN EQUATION

The confluent Heun differential equation

$$g''(t) + g'(t)\left(\zeta + \frac{\gamma}{t} + \frac{\delta}{t-1}\right) + \frac{g(t)(\beta t - q)}{(t-1)t} = 0 \quad (1)$$

is an ordinary differential equation of second order with two regular singularities at $t = 0, 1$ and an irregular one at infinity. A basis of solutions is given by

$$\text{HeunC}[q, \beta, \gamma, \delta, \zeta, t] \quad (2)$$
$$t^{1-\gamma}\text{HeunC}[(1-\gamma)(\zeta-\delta)+q, \beta+(1-\gamma)\zeta, 2-\gamma, \delta, \zeta, t]$$

with HeunC the confluent Heun function. It is convenient to bring the irregular singularity to the origin, by taking $t = 1/z$, and introduce the parameters $u$, $x$, $\kappa_i$ (with $i = 1, 2, 3$) for the confluent Heun labels

$$\beta = -x(\kappa_1 + \kappa_3), \quad \delta = \kappa_2 + \kappa_3, \quad \gamma = 1 - \kappa_2 + \kappa_3$$
$$\zeta = -x \quad , \quad q = u - \kappa_3^2 - x(\kappa_1 + \kappa_3) \quad (3)$$

In these variables, the confluent Heun equation (CHE) takes the form

$$G''(z)+G'(z)\left(\frac{x}{z^2}+\frac{2\kappa_1+1}{z}+\frac{\kappa_2+\kappa_3}{z-1}\right)+G(z)\frac{\left(u-\kappa_1^2+(\kappa_1+\kappa_2)(\kappa_1+\kappa_3)z\right)}{(z-1)z^2}=0 \qquad (4)$$

with

$$G(z)=z^{-\kappa_1-\kappa_3}g(\tfrac{1}{z}) \qquad (5)$$

There are two limits in which (4) admits a simple solution. They correspond to sending $x\to 0$ keeping finite either $z$ or the ratio

$$y=\frac{x}{z} \qquad (6)$$

In these limits (4) reduces to a hypergeometric equation with solutions

$$G_\alpha(z)\underset{\substack{x,z\to 0 \\ x/z \text{ finite}}}{\approx}H_\alpha^0(\tfrac{x}{z}) \quad , \quad G_\alpha(z)\underset{\substack{x\to 0 \\ z \text{ finite}}}{\approx}H_\alpha^1(z) \quad (7)$$

where $\alpha=\pm$ labels the two independent solutions,

$$\begin{aligned}H_\alpha^0(y)&=y^{-\alpha a+\kappa_1}{}_1F_1(\kappa_1-\alpha a,1-2\alpha a;y)\\ H_\alpha^1(z)&=z^{\alpha a-\kappa_1}{}_2F_1(\alpha a+\kappa_2,\alpha a+\kappa_3;1+2\alpha a;z)\end{aligned} \quad (8)$$

and

$$a=\sqrt{u}+O(x) \qquad (9)$$

a non-trivial function of $u$ to be determined. We will look for solutions in the domain $z\in[0,1]$. The general $G(z)=\sum_\alpha c_\alpha G_\alpha(z)$ behaves near the boundaries as

$$\begin{aligned}G(z)&\underset{z\to 0}{\approx}B_-(1+\ldots)+B_+z^{1-2\kappa_1}e^{\frac{x}{z}}(1+\ldots)\\ G(z)&\underset{z\to 1}{\approx}D_-(1+\ldots)+D_+(1-z)^{1-\kappa_2-\kappa_3}(1+\ldots)(10)\end{aligned}$$

where the dots stand for subleading terms in these limits. Specifying boundary conditions corresponds to fix two linear combinations made out of the four coefficients $B_\pm$, $D_\pm$.

We notice that the two limits (7) overlap in the region where both $z$ and $y$ are small. We refer to this region as *the near zone* and here we look for solutions as a double series

$$\text{Near zone}: G_\alpha(z)=z^{\alpha a-\kappa_1}\sum_{i,j=0}^\infty G_{\alpha ij}(a)\,z^i\,(\tfrac{x}{z})^j \quad (11)$$

with $x\ll z\ll 1$. $a(u)$ given in (9). The quantity $a$ plays the role of a 'Floquet exponent', specifying the monodromy of $G_\alpha$ around a contour enclosing the points

0 and $x$. As we will see, the solutions of (4) and the connection formulae take a simpler form if written in terms of the variable $a$ rather than $u$. So we write

$$u=a^2+\sum_{i=1}^\infty u_i(a)x^i \qquad (12)$$

The coefficients $G_{\alpha ij}(a)$ and $u_i(a)$ can be determined recursively order by order in the variable $x$ and $a(u)$ follows by inverting the series (12).

## A. PM approximation: exterior region

In this subsection we consider the limit $x\ll 1$ keeping $y$ finite. This corresponds to the PM approximation and we refer to this region as *the exterior region*: the reason of it will be clear in the next Section from the dictionary with the gravity side (see (40)).
We denote the basic solution as $G_\alpha^0(\tfrac{x}{z})$ and at leading order in $x$ we take

$$G_\alpha^0(y)=H_\alpha^0(y) \qquad (13)$$

For higher orders in $x$, proceeding by trial and error, we come to the following ansätze

$$G_\alpha^0(y)=P_0(y)\,H_\alpha^0(y)+\widehat{P}_0(y)\,yH_\alpha^{0\prime}(y) \qquad (14)$$

where the prime stands for derivation and

$$P_0(y)=1+\sum_{i=1}^k\sum_{j=0}^{i-1}c_{ij}\,x^iy^{j-i},\quad \widehat{P}_0(y)=\sum_{i=1}^k\sum_{j=0}^{i-1}\widehat{c}_{ij}\,x^iy^{j-i} \qquad (15)$$

are polynomials of order $k$ in $x$[1]. We plug this ansätze into (4) and use the differential equation to express higher derivatives of $H_\alpha^0$ in terms of $H_\alpha^0$ and its first derivative themselves. Therefore one finds a system of algebraic linear equations that determines the coefficients $c_{ij}$, $\widehat{c}_{ij}$ and $u_i$ (whose complete forms are given in (67)). Setting $c_{00}=1$, one finds, for the first term in the recursion

$$\begin{aligned}c_{10}=u_1&=\frac{\left[2a^2\left(\kappa_1+\kappa_2+\kappa_3-\frac{1}{2}\right)+2\kappa_1\kappa_2\kappa_3-\kappa_1\kappa_2-\kappa_1\kappa_3-\kappa_2\kappa_3\right]}{4a^2-1}\\ \widehat{c}_{10}&=-\partial_{\kappa_1}u_1=-\frac{\left(2a^2+2\kappa_2\kappa_3-\kappa_2-\kappa_3\right)}{4a^2-1}\end{aligned} \qquad (16)$$

---

[1] For $k=1$ M.Billò, M.L.Frau and A.Lerda first obtained (14) resumming the contributions of the Young tableaux entering the partition function of the supersymmetric gauge theory.

It is easy to check that $c_{ij}$, $\widehat{c}_{ij}$ are invariant under $a \to -a$. This is true at any PM order, so all the dependence on $\alpha$ of the solution is codified in $H_\alpha^0$. The results (16) are special cases of more general relations that determine the coefficients $c_{i,i-1}$, $\widehat{c}_{i,i-1}$ in terms of $u$

$$c_{i,i-1} = u_i \,,$$

$$\gamma_\kappa = 1 + \sum_{i=1}^\infty \widehat{c}_{i,i-1} x^i = \exp\left\{-\sum_{i=1}^\infty \frac{\partial_{\kappa_1} u_i}{i} x^i\right\} \quad (17)$$

The combination in the exponential will be denoted in the following as

$$\mathcal{F}_{\text{inst}}(a, x) \equiv -\sum_{i=1}^\infty \frac{u_i(a)}{i} x^i \quad (18)$$

The subscript "inst" reminds the reader that this quantity is interpreted as the instanton prepotential in the gauge theory description of the Heun equation. Indeed in the limit $z \to 0$, where one of the two gauge couplings is turned off, the quiver theory reduces to a $\mathcal{N} = 2$ supersymmetric $SU(2)$ gauge theory with three massive fundamental flavours with prepotential $\mathcal{F}_{\text{inst}}$.

We notice that (14) is exact in $y$ at the k-th PM order, so we can extrapolate it to the region of large $y$ using the standard hypergeometric transformation rule

$$H_\alpha^0(y) = \sum_{\alpha'=\pm} B_{\alpha\alpha'} \widetilde{H}_{\alpha'}^0(y) \quad (19)$$

with

$$\widetilde{H}_\alpha^0(y) = y^{\frac{\alpha+1}{2}(2\kappa_1 - 1)} e^{\frac{y(1+\alpha)}{2}} {}_2F_0\left(\tfrac{1+\alpha}{2} - \alpha\kappa_1 - a, \tfrac{1+\alpha}{2} - \alpha\kappa_1 + a; \alpha y^{-1}\right) \quad (20)$$

and

$$B_{\alpha\alpha'} = e^{\frac{i\pi(1-\alpha')}{2}(\kappa_1 - \alpha a)} \frac{\Gamma(1 - 2\alpha a)}{\Gamma\left(\frac{1-\alpha'}{2} - \alpha a + \alpha'\kappa_1\right)} \quad (21)$$

We can therefore introduce an equivalent basis of solutions, naturally defined in the *far region*.

$$\widetilde{G}_\alpha^0(y) = P_0(y) \widetilde{H}_\alpha^0(y) + \widehat{P}_0(y) y \widetilde{H}_\alpha^{0\prime}(y) \quad (22)$$

related to the previous one by the connection formulae

$$G_\alpha^0(y) = \sum_{\alpha'=\pm} B_{\alpha\alpha'} \widetilde{G}_{\alpha'}^0(y) \quad (23)$$

### B. PN approximation: interior region

In this subsection we consider the limit $x \ll 1$ keeping $z$ finite. This corresponds to the PN approximation and we refer to this region as *the interior region*. The analysis proceeds *mutatis mutandis* following the same steps of the previous subsection, exchanging the roles of $y$ and $z$. We denote by $G_\alpha^1(z)$ the basic solutions and at leading order in $x$ we take

$$G_\alpha^1(z) = H_\alpha^1(z) \quad (24)$$

Higher orders follow from the ansätze

$$G_\alpha^1(z) = P_1(z) H_\alpha^1(z) + \widehat{P}_1(z) z H_\alpha^{1\prime}(z) \quad (25)$$

where the prime stands for derivation and

$$P_1(z) = 1 + \sum_{j=1}^k \sum_{i=0}^{j-1} d_{ij} z^{i-j} x^j$$

$$\widehat{P}_1(z) = \sum_{j=1}^k \sum_{i=0}^j \widehat{d}_{ij} z^{i-j} x^j \quad (26)$$

Setting $d_{00} = 1$, one finds that the first non-trivial coefficients are

$$d_{01} = \frac{2\left(a^2 - \kappa_1^2\right)}{4a^2 - 1}, \quad \widehat{d}_{11} = -\widehat{d}_{01} = \frac{(2\kappa_1 - 1)}{4a^2 - 1} \quad (27)$$

We notice that the results are invariant under $a \to -a$. This is true at any PN order, so all the dependence on $\alpha$ of the solution is codified in $H_\alpha^1$.

Proceeding as before, we can extrapolate the PN representation to the region $z \approx 1$ using the hypergeometric transformation rule

$$H_\alpha^1(z) = \sum_{\alpha'=\pm} F_{\alpha\alpha'} \widetilde{H}_{\alpha'}^1(z) \quad (28)$$

with

$$\widetilde{H}_\alpha^1(z) = z^{a-\kappa_1}(1-z)^{\frac{1+\alpha}{2}(1-\kappa_2-\kappa_3)} {}_2F_1\left(\tfrac{1+\alpha}{2} - \alpha\kappa_2 + a, \tfrac{1+\alpha}{2} - \alpha\kappa_3 + a; 1 + \alpha\left(1 - \kappa_2 - \kappa_3\right); 1 - z\right)$$

and

$$F_{\alpha\alpha'} = \frac{\Gamma(1 + 2\alpha a)\Gamma(\alpha'[\kappa_2 + \kappa_3 - 1])}{\Gamma\left(\frac{1-\alpha'}{2} + \alpha a + \alpha'\kappa_2\right)\Gamma\left(\frac{1-\alpha'}{2} + \alpha a + \alpha'\kappa_3\right)} \quad (29)$$

An equivalent basis of solutions defined in the *near hori-*

*zon region* is therefore given by

$$\widetilde{G}^1_\alpha(z) = P_1(z)\,\widetilde{H}^1_\alpha(z) + \widehat{P}_1(z)\,z\,\widetilde{H}^{1\prime}_\alpha(z) \qquad (30)$$

satisfying the connection formulae

$$G^1_\alpha(z) = \sum_{\alpha'=\pm} F_{\alpha\alpha'}\widetilde{G}^1_{\alpha'}(z) \qquad (31)$$

### C.  Boundary conditions and connection formulae

The functions $G^0_\alpha(\frac{x}{z})$ and $G^1_\alpha(z)$ are solutions of the same differential equation and have the same monodromy around $z = 0$, so they should be proportional to each

other. We denote by $g_\alpha(x)$ their ratio

$$g_\alpha(x) = \frac{G^1_\alpha(z)}{G^0_\alpha(\frac{x}{z})} \qquad (32)$$

Expanding for small $z$ and $y$ one can check explicitly that this ratio depends only on $x = zy$. Furthermore one finds

$$\gamma(x) = \frac{g_+(x)}{g_-(x)} = x^{2a}e^{-\partial_a\mathcal{F}_{\rm inst}(a,x)} \qquad (33)$$

with $\mathcal{F}_{\rm inst}(a,x)$ defined in (18). Collecting all the different pieces together, we have at our disposal four different bases of solutions, so that the general solution can be written as

$$
\begin{aligned}
G(z) &= \sum_\alpha c_\alpha g_\alpha(x) G^0_\alpha(\tfrac{x}{z}) + \ldots = \sum_{\alpha,\alpha'} c_\alpha g_\alpha(x) B_{\alpha\alpha'}\widetilde{G}^0_{\alpha'}(\tfrac{x}{z}) + \ldots \\
&= \sum_\alpha c_\alpha G^1_\alpha(z) + \ldots = \sum_{\alpha,\alpha'} c_\alpha F_{\alpha\alpha'}\widetilde{G}^1_{\alpha'}(z) + \ldots
\end{aligned}
\qquad (34)
$$

The first line corresponds to the PM approximation where $x$ is taken small keeping $y$ finite. The second equality in this line is obtained using the connection formulae (23). Similarly, the second line corresponds to the PN approximation where $x$ is taken small keeping $z$ finite

and the right hand side follows from (31). Finally dots stand for terms of order $x^{k+1}$ in the corresponding limits. Expanding at the leading order the two right hand sides in (34) and using (17), one finds the asymptotics

$$
\begin{aligned}
G(z) &\underset{z\to 0}{\approx} \sum_{\alpha,\alpha'} c_\alpha g_\alpha(x) B_{\alpha\alpha'}(\tfrac{x}{z})^{\frac{1+\alpha'}{2}(2\kappa_1-1)} e^{\frac{x(1+\alpha')}{2z}} e^{\frac{1+\alpha'}{2}\partial_{\kappa_1}\mathcal{F}_{\rm inst}} \\
G(z) &\underset{z\to 1}{\approx} \sum_{\alpha,\alpha'} c_\alpha F_{\alpha\alpha'}(1-z)^{\frac{1+\alpha'}{2}(1-\kappa_2-\kappa_3)} h_{\alpha'}
\end{aligned}
\qquad (35)
$$

with

$$h_{\alpha'} = P_1(1) + \frac{1+\alpha'}{2}(1-\kappa_2-\kappa_3)\widehat{P}_1{}'(1) \qquad (36)$$

Once again the ratio of these coefficients takes a simple form

$$\frac{h_+}{h_-} = e^{-\partial_{\kappa_2}+\kappa_3\mathcal{F}_{\rm inst}} \qquad (37)$$

Comparing against (10) one finds the asymptotic coefficients

$$
\begin{aligned}
B_{\alpha'} &= \sum_\alpha c_\alpha g_\alpha(x) B_{\alpha\alpha'} x^{\frac{1+\alpha'}{2}(2\kappa_1-1)} e^{\frac{1+\alpha'}{2}\partial_{\kappa_1}\mathcal{F}_{\rm inst}} \\
D_{\alpha'} &= \sum_\alpha c_\alpha F_{\alpha\alpha'} h_{\alpha'}
\end{aligned}
\qquad (38)
$$

## III.  BLACK HOLE PERTURBATION

In this Section we apply the results obtained in the previous one to the study of gravitational waves in the Schwarzschild geometry.

### A.  Heun gravity correspondence

BHs perturbations are typically described by equations of the confluent Heun type. For example, the Teukolsky equation for spin $s$ perturbations in the Schwarzschild geometry is given by

$$\frac{1}{\Delta(r)^s}\frac{d}{dr}\left[\Delta(r)^{s+1}R'(r)\right] + \left(\frac{\omega^2 r^4 - 2is(r-M)\omega r^2}{\Delta(r)} + 4is\omega r - (\ell-s)(\ell+s+1)\right)R(r) = 0 \tag{39}$$

with $\Delta = r(r-2M)$. We are interested in the $\psi_4$ fundamental mode [8, 55] corresponding to $s = -2$. (39) can

be put into the CHE form (4) after the identifications (look at Appendix A for details)

$$R(z) = e^{-\frac{x}{2z}}(1-z)^{\frac{1+\kappa_2+\kappa_3}{2}}\left(\frac{x}{z}\right)^{\frac{3}{2}-\kappa_1}G(z) \quad , \quad z = \frac{2M}{r} \quad , \quad x = 4iM\omega \tag{40}$$

$$\kappa_1 = \kappa_2^* = \tfrac{5}{2} + 2iM\omega \quad , \quad \kappa_3 = \tfrac{1}{2} - 2iM\omega \quad , \quad u = (\ell+\tfrac{1}{2})^2 + 10iM\omega - 4\omega^2 M^2$$

We denote by $R_{\rm in}(z)$ a solution satisfying incoming boundary conditions at the horizon and by $R_{\rm up}(z)$ that

one satisfying upgoing boundary conditions at infinity, i.e.

$$R_{\rm in}(z) \approx \begin{cases} B_-^{\rm in}e^{-\frac{x}{2z}}z^{\kappa_1-\frac{3}{2}} + B_+^{\rm in}e^{\frac{x}{2z}}z^{-\kappa_1-\frac{1}{2}} & z \to 0 \\ D_-^{\rm in}(1-z)^{\frac{1+\kappa_2+\kappa_3}{2}} & z \to 1 \end{cases}$$

$$R_{\rm up}(z) \approx \begin{cases} B_+^{\rm up}e^{\frac{x}{2z}}z^{-\kappa_1-\frac{1}{2}} & z \to 0 \\ D_-^{\rm up}(1-z)^{\frac{1+\kappa_2+\kappa_3}{2}} + D_+^{\rm up}(1-z)^{\frac{3-\kappa_2-\kappa_3}{2}} & z \to 1 \end{cases} \tag{41}$$

where we have set to zero the coefficients $D_+^{\rm in}$ and $B_-^{\rm up}$. In radial coordinates $R_{\rm in,up}(r) \equiv R_{\rm in,up}[z(r)]$. In terms of these solutions the Green function reads

$$G(r,r') = \frac{1}{W}\begin{cases} R_{\rm in}(r')R_{\rm up}(r) & r' < r \\ R_{\rm in}(r)R_{\rm up}(r') & r < r' \end{cases} \tag{42}$$

with

$$W = \frac{R_{\rm in}(r)R'_{\rm up}(r) - R'_{\rm in}(r)R_{\rm up}(r)}{\Delta(r)} = \frac{i\omega}{2M^2}B_-^{\rm in}B_+^{\rm up} \tag{43}$$

a constant built out of the Wronskian which is conveniently computed for $r \to \infty$, since its first derivative is zero as it follows using (39). In the following we use as independent variables the two dimensionless combinations

$$y = 2i\omega r \quad , \quad x = 4i\omega M \tag{44}$$

The incoming and upgoing solutions can be written as linear combinations

$$R_{\rm in,up}(y) = \sum_\alpha c_\alpha^{\rm in,up} g_\alpha R_\alpha(y) \tag{45}$$

of the fundamental solutions $R_\alpha(y)$. Since we are mainly interested in the physics at large distances $z \ll 1$, it is convenient to work with the PM representation and write the two independent solutions of the Teukolsky equation as

$$R_\alpha(y) = e^{-\frac{y}{2}}(1-\tfrac{x}{y})^{2-\frac{x}{2}}y^{-1-\frac{x}{2}}G_\alpha^0(y) \tag{46}$$

$c_\alpha^{\rm in,up}$ are some coefficients, chosen such that the incoming boundary conditions at the horizon or upgoing at infinity are satisfied, i.e.

$$D_+^{\rm in} = B_-^{\rm up} = 0 \tag{47}$$

with asymptotic coefficients given by (10), (40)

$$B_{\alpha'}^{\rm in,up} = \sum_\alpha c_\alpha^{\rm in,up}g_\alpha(x)B_{\alpha\alpha'}x^{\alpha'\kappa_1+1-\frac{\alpha'}{2}}e^{\frac{1+\alpha'}{2}\partial_{\kappa_1}\mathcal{F}_{\rm inst}}$$

$$D_{\alpha'}^{\rm in,up} = \sum_\alpha c_\alpha^{\rm in,up}F_{\alpha\alpha'}h_{\alpha'}e^{-\frac{x}{2}}x^{\frac{3}{2}-\kappa_1} \tag{48}$$

We are interested in the PN expansion of the solutions, since we want to describe binary systems in the limit where the distance between the two objects is large and velocities small. For example, for circular orbits with tangential velocity $v$, this corresponds to the limit where $v$ is small, $y \sim v$ and $x \sim v^3$. Therefore the PN expansion can be written as

$$R_\alpha(y) = y^{\frac{3}{2}-\alpha a}\sum_{n=0}^\infty R_n(\alpha a, y) \tag{49}$$

with

$$R_0(a, y) = 1$$

$$R_1(a, y) = \frac{2y}{1 - 2a}$$

$$R_2(a, y) = \frac{(17 - 2a)y^2}{16(2a - 1)(a - 1)} + \frac{(2a - 3)x}{4y}$$

$$R_3(a, y) = \frac{(7 - 2a)y^3}{8(2a - 1)(2a - 3)(a - 1)} + \frac{(11 - 6a)x}{4(2a - 1)}$$

$$R_4(a, y) = \frac{\left(4a^2 - 72a + 163\right) y^4}{512(a-2)(a-1)(2a-3)(2a-1)}$$
$$+ \frac{\left(-4a^2 + 48a - 119\right) yx}{64(2a-1)(a-1)} + \frac{\left(8a^3 - 4a^2 - 18a + 9\right) x^2}{64y^2(a+1)} \quad (50)$$

and

$$a(\ell) = \ell + \frac{1}{2} + \frac{\left(15\ell^4 + 30\ell^3 + 28\ell^2 + 13\ell + 24\right) x^2}{8\ell(\ell + 1)(2\ell + 1)\left(4\ell^2 + 4\ell - 3\right)} + \dots \quad (51)$$

## B. The incoming solution

The incoming solution is obtained by requiring no outgoing waves at the horizon, i.e. by setting $D_+^{\rm in} = 0$. According to (48), such condition leads to

$$\sum_{\alpha=\pm} c_\alpha^{\rm in} F_{\alpha+} = 0 \quad (52)$$

This determines the ratio of the two coefficients to be

$$\frac{c_+^{\rm in}}{c_-^{\rm in}} = -\frac{F_{-+}}{F_{++}} = \frac{\Gamma(-2a)\Gamma(\kappa_2+a)\Gamma(\kappa_3+a)}{\Gamma(2a)\Gamma(\kappa_2-a)\Gamma(\kappa_3-a)} \quad (53)$$

On the other hand, using the first of (35), one finds the asymptotic behaviour at infinity

$$R_{\rm in}(y) \underset{y\to\infty}{\approx} \sum_{\alpha,\alpha'} c_\alpha^{\rm in} g_\alpha B_{\alpha\alpha'} e^{\frac{\alpha' y}{2}} y^{\alpha'\kappa_1 + 1 - \frac{\alpha'}{2}} e^{\frac{1+\alpha'}{2}\partial_{\kappa_1}} \mathcal{F}_{\rm inst} \quad (54)$$

Comparing against the first line of $R_{\rm in}$ in (41), one finds

$$B_{\alpha'}^{\rm in} = \sum_\alpha c_\alpha^{\rm in} g_\alpha B_{\alpha\alpha'} x^{\alpha'\kappa_1 + 1 - \frac{\alpha'}{2}} e^{\frac{1+\alpha'}{2}\partial_{\kappa_1}} \mathcal{F}_{\rm inst} \quad (55)$$

We notice that the incoming boundary condition determines only the ratio $c_+^{\rm in}/c_-^{\rm in}$, leaving undetermined the overall normalization $c_-^{\rm in}$. This dependence however cancels out in the Green function after dividing by the factor $W$ (43). It is convenient therefore to define a ratio which does not depend on overall normalizations

$$\mathfrak{R}_{\rm in}(y) = \frac{R_{\rm in}(y)}{x B_-^{\rm in}} = C_{\rm in}\left[R_-(y) - \gamma \frac{F_{-+}}{F_{++}} R_+(y)\right] \quad (56)$$

with

$$C_{\rm in} = \frac{x^{\frac{x}{2}}}{B_{--}}\left(1 - \gamma\frac{F_{-+}B_{+-}}{F_{++}B_{--}}\right)^{-1} \underset{x\to 0}{\approx} (-1)^{\ell+1}\frac{\Gamma(\ell-1)}{\Gamma(2\ell+2)} + \dots \quad (57)$$

We notice that in the PN limit $R_\alpha(y) \sim y^{\frac{3}{2} - \alpha a}$ and $\gamma \sim x^{2a}$, so the $\gamma R_+$-contribution in (56) is suppressed by an extra factor $(x/y)^{2a}$ with respect to $R_-$. The incoming solution is therefore dominated by the $R_-$ component. Using (51) one finds the PN expansion

$$R_-(y) = y^{a+\frac{3}{2}}\left[1 + \frac{y}{1+\ell} + \left(\frac{(9+\ell)y^2}{8(1+\ell)(3+2\ell)} - \frac{(\ell+2)x}{2y}\right) + \left(\frac{(\ell+4)y^3}{8(\ell+1)(\ell+2)(2\ell+3)} - \frac{(7+3\ell)x}{4(\ell+1)}\right)\right.$$
$$\left. + \left(\frac{(50+19\ell+\ell^2)y^4}{128(\ell+1)(\ell+2)(2\ell+3)(2\ell+5)} + \frac{(\ell^2-1)(\ell+2)x^2}{4(2\ell-1)y^2} - \frac{(\ell+4)(\ell+9)xy}{16(\ell+1)(2\ell+3)}\right) + \dots\right] \quad (58)$$

It is important to observe that at higher orders both $R_\pm(y)$ exhibit poles in the limit where $\ell \to \mathbb{Z}$, but one can check that they cancel against each other in the combination (56). For example, for $\ell = 2$ the poles in the two components first appear at order $v^{16}$.

## C. The upgoing solution

Upgoing boundary conditions correspond to setting $B_-^{\rm in} = 0$, that according to (48) leads to

$$\frac{c_-^{\rm up}}{c_+^{\rm up}} = -\gamma\frac{B_{+-}}{B_{--}} \quad (59)$$

Following the same passages of the incoming solution, we can write the asymptotic behaviour of $R_{\rm up}(y)$ at infinity, which is the same as (54) but with $c_\alpha^{\rm up}$ in place of $c_\alpha^{\rm in}$. This implies that also the coefficients $B_\alpha^{\rm up}$ take the same form of (55) with the same substitution $c_\alpha^{\rm in} \to c_\alpha^{\rm up}$. Again

we introduce the ratio

$$\mathfrak{R}_{\mathrm{up}}(y) = (2M)^3 \frac{R_{\mathrm{up}}(y)}{B_+^{\mathrm{up}}} = C_{\mathrm{up}} \left[ R_+(y) - \tfrac{B_{+-}}{B_{--}} R_-(y) \right] \tag{60}$$

with

$$\begin{aligned}
C_{\mathrm{up}} &= \frac{(2M)^3 e^{-\partial_{\kappa_1}\mathcal{F}_{\mathrm{inst}}}}{x^{3+\frac{x}{2}} B_{++}} \left(1 - \tfrac{B_{-+}B_{+-}}{B_{++}B_{--}}\right)^{-1} = \\
&= \frac{(2M)^3}{x^{3+\frac{x}{2}}} \frac{\Gamma(2-\ell)}{\Gamma(-2\ell)} + \cdots
\end{aligned} \tag{61}$$

We notice that the leading term on the right hand side is finite in the limit where $\ell$ becomes an integer with $\ell \geq 2$. The same result is obtained by setting $\ell \to \mathbb{N}$ from the very beginning and sending then $x \to 0$. In the PN limit, the solution is dominated by the $R_+(y)$ component with PN expansion

$$\begin{aligned}
R_+(y) = y^{\frac{3}{2}-a} &\left[ 1 - \frac{y}{\ell} + \left( \frac{(\ell-1)x}{2y} - \frac{(\ell-8)y^2}{8l(2\ell-1)} \right) + \left( -\frac{(3\ell-4)x}{4\ell} + \frac{(\ell-3)y^3}{8(\ell-1)\ell(2\ell-1)} \right) \right. \\
&\left. + \left( \frac{(\ell-1)\ell(\ell+2)x^2}{4(2\ell+3)y^2} - \frac{(\ell-8)(\ell-3)xy}{16\ell(2\ell-1)} + \frac{(32-17\ell+\ell^2)\,y^4}{128(\ell-1)\ell(2\ell-3)(2\ell-1)} \right) \right]
\end{aligned} \tag{62}$$

It is important to observe that again the poles in $R_{\pm}(y)$ in the limit where $\ell \to \mathbb{Z}$ cancel against each other in the combination (60). For $\ell = 2$ the poles in the two components first appear at order $v^7$.

## IV. TESTS AGAINST MST AND NUMERICAL SIMULATIONS

In this Section we test our hypergeometric PM representation against the MST method [10–19] and numerical simulations. When we restrict ourselves to circular orbits[2], we work at order 30PN, i.e. $v^{60}$, with $v$ the tangential velocity of the probe particle. We use the package *Teukolsky* of the Black Hole Perturbation Toolkit (BHPT) [56] both for the computations using the MST method and for numerical integration of the differential equation. The observables we consider are the $(\ell, m)$ harmonic modes of incoming solution $\mathfrak{R}_{\mathrm{in}}$ and the energy flux radiated to infinity by the system. The results for the energy flux are also compared against the 22PN results in [20].

The numerical computations and those with the MST method are performed setting in the BHPT Mathematica files: AccuracyGoal = 1000, PrecisionGoal = 70, WorkingPrecision = 80, SetPrecision=80. With this choice of parameters both numerical and MST results agree to a 15 digits precision. We restrict our comparisons to circular orbits for the aforementioned observables, so $r$ should be taken bigger than the innermost stable circular orbit

(ISCO) radius $r_{\mathrm{ISCO}} = 6M$. Consequently the domain of variability of the tangential velocity is $v \in [0, \sqrt{\frac{M}{r}} = \frac{1}{\sqrt{6}}]$.

### A. The incoming and upgoing solutions

In this subsection we collect all the ingredients needed in the computations of the incoming, upgoing waveforms at order k=30 in the hypergeometric PM representation. We write[3]

$$\begin{aligned}
\mathfrak{R}_{\mathrm{in}}(y) &= C_{\mathrm{in}} R_-(y) + (a \to -a) \\
\mathfrak{R}_{\mathrm{up}}(y) &= C_{\mathrm{up}} R_+(y) + (a \to -a)
\end{aligned} \tag{63}$$

with

$$\begin{aligned}
R_\alpha(y) &= e^{-\frac{y}{2}}(1 - \tfrac{x}{y})^{2-\frac{x}{2}} y^{-1-\frac{x}{2}} \left[ P_0(y) H_\alpha^0(y) + \widehat{P}_0(y)\, y H_\alpha^{0\prime}(y) \right] \\
H_\alpha^0(y) &= y^{\frac{5}{2}+\frac{x}{2}-\alpha a} {}_1F_1\left(\tfrac{5}{2} + \tfrac{x}{2} - \alpha a; 1 - 2\alpha a; y\right) \\
C_{\mathrm{up}} &= \frac{(2M)^3}{x^{3+\frac{x}{2}} \gamma_\kappa} \frac{\Gamma(\frac{5}{2} - a + \frac{x}{2})}{\Gamma(1 - 2a)} \left(1 - e^{-2\pi i a} \frac{\cos\frac{\pi}{2}(x+2a)}{\cos\frac{\pi}{2}(x-2a)}\right)^{-1} \\
\gamma_\kappa &= 1 + \sum_{i=1}^{\infty} \widehat{c}_{i,i-1} x^i \\
C_{\mathrm{in}} &= x^{\frac{x}{2}} e^{-\frac{i\pi}{2}(5+2a+x)} \frac{\Gamma(-\frac{3}{2} + a - \frac{x}{2})}{\Gamma(1 + 2a)} (1 - \gamma_0\,\gamma)^{-1}
\end{aligned} \tag{64}$$

---

[2] As it is well known in this case the PM and PN expansion are just one expansion.

[3] Here we rewrite the second terms in the right hand sides of (56), (60), as the images under $a \to -a$ of the first terms, using the transformation rules $R_- \to R_+$, $\gamma \to \gamma^{-1}$, $F_{-\alpha'} \leftrightarrow F_{+\alpha'}$ and $B_{-\alpha'} \leftrightarrow B_{+\alpha'}$.

and[4]

$$\gamma_{\text{tot}} = \underbrace{e^{-2\pi i a} \frac{\Gamma(-2a)^2\Gamma\left(-\frac{3}{2}+a-\frac{x}{2}\right)\Gamma\left(\frac{1}{2}+a-\frac{x}{2}\right)\Gamma\left(\frac{5}{2}+a-\frac{x}{2}\right)}{\Gamma(2a)^2\Gamma\left(-\frac{3}{2}-a-\frac{x}{2}\right)\Gamma\left(\frac{1}{2}-a-\frac{x}{2}\right)\Gamma\left(\frac{5}{2}-a-\frac{x}{2}\right)}}_{\gamma_0 = \frac{F_{-+}B_{+-}}{F_{++}B_{--}}} \underbrace{x^{2a}e^{-\partial_a\mathcal{F}_{\text{inst}}}}_{\gamma} \;,\; \mathcal{F}_{\text{inst}} = -\sum_{i=1}^{\infty}\frac{u_i}{i}x^i \tag{65}$$

The coefficients $c_{ij}, \widehat{c}_{ij}$ in the polynomials

are determined by the recursion relations

$$P_0(y) = 1 + \sum_{i=1}^{k}\sum_{j=0}^{i-1} c_{ij}\, x^i y^{j-i} \;,\; \widehat{P}_0(y) = \sum_{i=1}^{k}\sum_{j=0}^{i-1}\widehat{c}_{ij}\, x^i y^{j-i} \tag{66}$$

$$
\begin{aligned}
c_{ij} &= \Big[ A_1\, c_{i,j-1} + A_2\, c_{i-1,j} + A_3\, c_{i-1,j-1} + \widehat{A}_1\widehat{c}_{i,j-1} + \widehat{A}_2\widehat{c}_{i-1,j} + \widehat{A}_3\widehat{c}_{i-1,j-1} \\
&\quad + \sum_{s=1}^{k} c_{s,s-1}(C_1\, c_{i-s,j-s} + \widehat{C}_1\,\widehat{c}_{i-s,j-s}) \Big]\left[8\Delta_{ij}(\Delta_{ij}^2 - 4a^2)\right]^{-1} \\
\widehat{c}_{ij} &= \Big[ B_1\, c_{i,j-1} + B_2\, c_{i-1,j} + B_3\, c_{i-1,j-1} + \widehat{B}_1\widehat{c}_{i,j-1} + \widehat{B}_2\widehat{c}_{i-1,j} + \widehat{B}_3\widehat{c}_{i-1,j-1} \\
&\quad + \sum_{s=1}^{k} c_{s,s-1}(C_2\, c_{i-s,j-s} + \widehat{C}_2\,\widehat{c}_{i-s,j-s}) \Big]\left[8\Delta_{ij}(\Delta_{ij}^2 - 4a^2)\right]^{-1} \\
u_i &= c_{i,i-1}
\end{aligned}
\tag{67}
$$

starting from $c_{0i} = \delta_{i0}$, $\widehat{c}_{0i} = 0$, with

$$
\begin{aligned}
A_1 &= -8\left(\Delta_{ij}+1\right)\left(\Delta_{ij}-x-5\right) \;,\; \widehat{A}_1 = 4\left(4a^2-(x+5)^2\right)\Delta_{i,j} \\
A_3 &= 8\Delta_{ij}\left(\Delta_{ij}-x-5\right) \;,\; \widehat{A}_3 = 4\left(4a^2-(x+5)^2\right)\left(-\Delta_{i,j}+x-2\right) \\
A_2 &= 2\left(3x^2+6x-12a^2-13\right)\Delta_{ij}+8\Delta_{ij}^3-8(x-1)\Delta_{ij}^2+8a^2(x-7)-2(x^3+3x^2-13x-15), \\
\widehat{A}_2 &= \left(4a^2-(x+5)^2\right)\left(-2(x-1)\Delta_{ij}+4a^2+x^2-2x-3\right) \\
B_1 &= -16\left(\Delta_{ij}+1\right) \;,\; \widehat{B}_1 = 8\Delta_{ij}\left(\Delta_{ij}+x+5\right) \;,\; \widehat{B}_3 = 8\left(-\Delta_{i,j}^2-7\Delta_{ij}+x^2+3x-10\right) \\
B_2 &= 4\left(-2(x-1)\Delta_{i,j}+4a^2+x^2-2x-3\right) \;,\; B_3 = 16\Delta_{ij} \\
\widehat{B}_2 &= 2\left(-\Delta_{ij}+x+1\right)\left(-4\Delta_{ij}^2-8\Delta_{ij}+12a^2+x^2+2x-15\right) \\
C_1 &= 8\left(\Delta_{ij}-x-5\right) \;,\; \widehat{C}_1 = 16a^2-4(x+5)^2 \;,\; C_2 = 16 \;,\; \widehat{C}_2 = 8\left(\Delta_{ij}+x+5\right)
\end{aligned}
\tag{68}
$$

and $\Delta_{ij} = i-j$.

Finally one has to compute $a(\ell)$. Given $u_i(a)$ by (67), one can invert the series (12) to compute $a(u)$ and then set (from (40))

$$u = (\ell + \tfrac{1}{2})^2 + \tfrac{5x}{2} + \tfrac{x^2}{4} \tag{69}$$

Inverting the series at high PM order can be time con-

suming for a Mathematica code. There is an alternative, more direct way to find the function $a(u)$ based on the connection between this quantity and the quantum SW period of a $SU(2)$ theory with three fundamental flavours. The quantum SW period $a$ is known to satisfy the continuous fraction equation [57]

---

[4] We notice that $\gamma_{\text{tot}} = e^{-\partial_a(\mathcal{F}_{\text{tree}}+\mathcal{F}_{\text{one-loop}}+\mathcal{F}_{\text{inst}})}$ collects the tree level, one-loop and instanton contributions to the prepoten-

tial of the underlying $SU(2)$ gauge theory with three fundamental flavours.

$$\frac{xM(a+1)}{P(a+1)-\frac{xM(a+2)}{P(a+2)-...}}+\frac{xM(a)}{P(a-1)-\frac{xM(a-1)}{P(a-2)-...}}-P(a)=0 \tag{70}$$

with

$$P(a) = a^2 + ax + 2x - \frac{3x^2}{4}x - (\ell+\tfrac{1}{2})^2,$$
$$M(a) = (a-\tfrac{x}{2}-\tfrac{1}{2})(a-\tfrac{x}{2}+\tfrac{3}{2})(a+\tfrac{x}{2}+\tfrac{3}{2}) \tag{71}$$

Equation (70) can be solved for $a(\ell)$ order by order in $x$. For example for $\ell = 2, 3$ one finds

$$a(2) = \tfrac{5}{2} + \tfrac{107x^2}{840} - \tfrac{1695233x^4}{148176000} + \tfrac{76720109901233x^6}{30764716012800000} - \tfrac{71638806585865707261481x^8}{99644321084605000704000000} + \cdots$$
$$a(3) = \tfrac{7}{2} + \tfrac{13x^2}{168} - \tfrac{10921x^4}{4346496} + \tfrac{95353832269x^6}{493385539046400} - \tfrac{23627105510827613x^8}{1148838359958761472000} + \cdots \tag{72}$$

### B. The luminosity

In this subsection, we collect the formulae needed to evaluate the luminosity at infinity for the case of a particle moving along a circular orbit in a Schwarzschild geometry.

After expanding in harmonics, the corresponding waveform at large distances can be written as

$$h = h_+ - ih_\times \underset{r\to\infty}{\approx} -\frac{4G}{r}\int\frac{d\omega}{2\pi}\sum_{\ell,m}e^{-i\omega(T-R_*)}\frac{Z_{\ell m}(\omega)}{2(i\omega)^2}Y_{-2}^{\ell m}(\theta,\phi)\,, \tag{73}$$

The harmonic coefficients $Z_{\ell m}$ can be written in terms of derivatives of the incoming solution

$$Z_{\ell m} = \Re_{\mathrm{in}}(y)\left(\frac{Ex^2 b_{0\ell m}}{2M^2 y(x-y)}+\frac{iJx^3\left(y^2+4y-4x\right)b_{1\ell m}}{8M^3 y^3(x-y)}+\frac{J^2 x^4\left(2x-4y-y^2\right)b_{2\ell m}}{64EM^4 y^3(x-y)}\right) \tag{74}$$
$$+y\partial_y\Re_{\mathrm{in}}(y)\left(\frac{iJx^3 b_{1\ell m}}{4M^3 y^3}+\frac{J^2 x^4\left(2x-2y-2y^2\right)b_{2\ell m}}{16EM^4 y^5}\right)-y^2\partial_y^2\Re_{\mathrm{in}}(y)\frac{J^2 x^4(x-y)b_{2\ell m}}{16EM^4 y^5}$$

with

$$b_{\ell m}^0 = \frac{\pi}{2}\sqrt{(\ell-1)\ell(\ell+1)(\ell+2)}\,Y_0^{\ell m}(\tfrac{\pi}{2},0),\ b_{\ell m}^1 = \pi\sqrt{(\ell-1)(\ell+2)}Y_{-1}^{\ell m}(\tfrac{\pi}{2},0),\ b_{\ell m}^2 = \pi\,Y_{-2}^{\ell m}(\tfrac{\pi}{2},0), \tag{75}$$

and

$$_sY_{\ell m}(\theta,\phi) = e^{im\phi}\sin^{2\ell}\left(\frac{\theta}{2}\right)\sqrt{\frac{(2\ell+1)(\ell-m)!(\ell+m)!}{4\pi(\ell-s)!(\ell+s)!}}\sum_{r=0}^{\ell-s}(-)^{\ell+m-r-s}\binom{\ell-s}{s}\binom{\ell+s}{r+s-m}\cot(\tfrac{\theta}{2})^{2r+s-m} \tag{76}$$

the spin-weighted spherical harmonics and

$$E = \frac{\mu(1-2v^2)}{\sqrt{1-3v^2}}\quad,\quad \omega J = \frac{\mu m v^2}{\sqrt{1-3v^2}} \tag{77}$$

the energy and angular momentum of the light particle. Finally, the luminosity, i.e. the amount of energy per unit time emitted by the system towards infinity, is computed by the formula (see [16] and references therein)

$$\frac{d\mathcal{E}}{dt} = \mu^2\sum_{\ell=2}^{\infty}\sum_{m=1}^{\ell}\frac{|Z_{\ell m}|^2}{2\pi\omega^2} = \mu^2\sum_{\ell=2}^{\infty}\sum_{m=1}^{\ell}\eta_{\ell m} \tag{78}$$

where $\mu$ is the mass of the orbiting body. Rather than the luminosity, we will display its ratio against the luminosity computed in the Newton approximation

$$\left(\frac{d\mathcal{E}}{dt}\right)_N = \frac{32\mu^2 v^{10}}{5M^2} = \mu^2\,\eta_{22}^0 \qquad (79)$$

with $\eta_{22}^0$ the contribution of the leading harmonic mode for $v \to 0$.

### C.   Numerical tests

In this subsection we use our formulae computed at order 30PM and set $M = 1$. In Figure 1, we plot the real and imaginary parts of the incoming and outgoing solutions obtained with this method for $\ell = 2$, $\omega = 0.1$.

The results are compared in Table I against those obtained by the MST method and numerical integration using the BHPT [56]. In all the subsequent tables, the displayed digits are those ones that are insensible to the contribution of terms of order higher than $v^{60}$ in $P_0(y)$ and $\widehat{P}_0(y)$. The agreement is up to 10 and 14 digits even for small values of $r$, where the PM approximation is less reliable.

In Figure 2 we display the $v$-dependence of the luminosity for the $(\ell, m) = (2, 2), (3, 3), (2, 1)$ harmonic modes computed from (56) at order $k = 30$ in the PN approximation.

The convergence speed of the results is showed in Table II. The digits in bold style underline the agreement between computations based on the PM hypergeometric representations and MST/numerical results. As expected, the speed of convergence varies with $v$: for $v = 0.1$ an agreement on 12 digits is already reached at 10PM order, while for $v = 0.4$, contributions at order 30PM are needed to reach a precision on 9 digits.

Finally in Tables III and IV we compare our 30PN results for the energy loss against those obtained by MST/Numerical and by the 22PN computation in [20].

We notice that in the whole range of $v$ the results obtained with the 30PN hypergeometric representations for the total luminosity $d\mathcal{E}/dt/(\mu^2\eta_{2,2}^0)$ agree with those obtained by MST and numerical integration up to 11-14 digits.

### Mathematica codes

Two Mathematica notebooks are given as ancillary files:

- **RinRupdRdt.nb** reproduces the results of Tables I, III and IV at order $k = 30$. This code imports the data files P30.m, Pt30.m, gg30.m, gk30.m and a30.m, containing the polynomials $P_0(y)$, $\widehat{P}_0(y)$, the quantities $\gamma$, $\gamma_\kappa$ and the $a$ cycle.

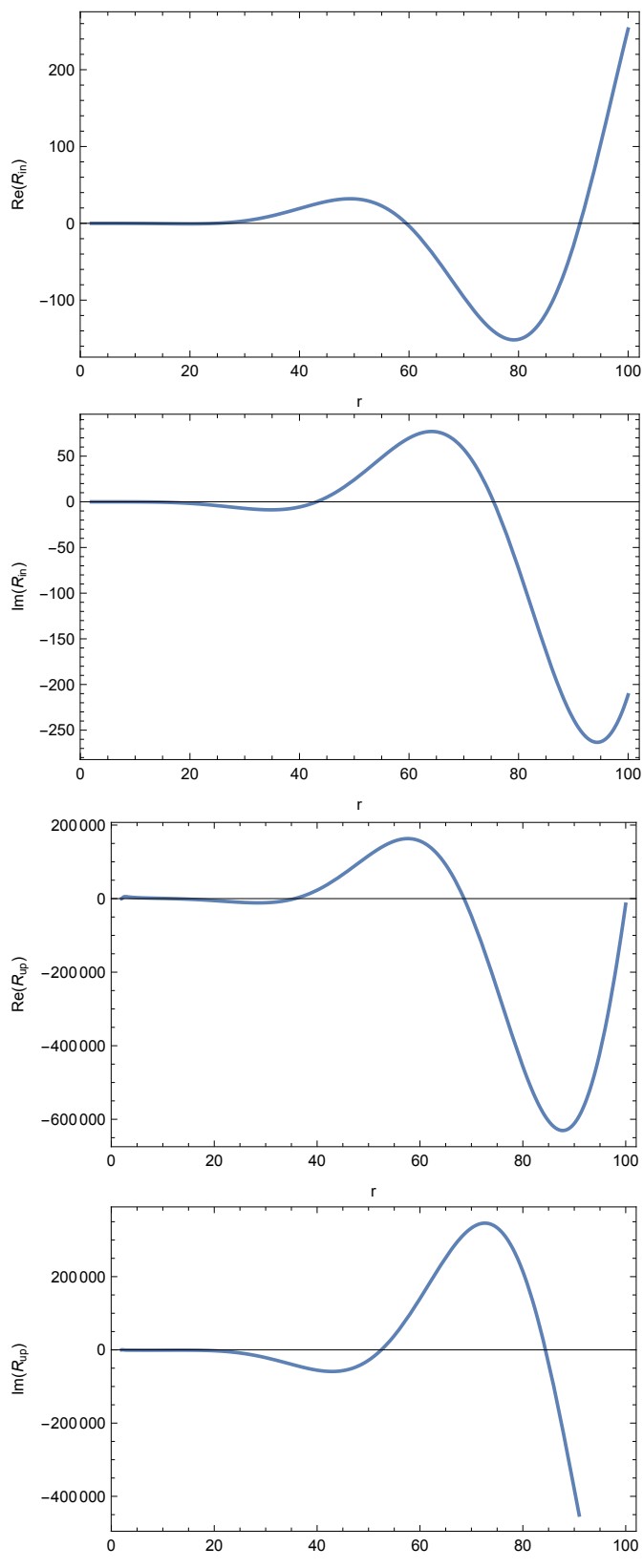

Figure 1: Plots of incoming and outgoing solutions for $\omega = 0.1, \ell = 2$. 1a: $\mathrm{Re}(\mathfrak{R}_{\mathrm{in}})$, 1b: $\mathrm{Im}(\mathfrak{R}_{\mathrm{in}})$, 2a: $\mathrm{Re}(\mathfrak{R}_{\mathrm{up}})$, 2b: $\mathrm{Im}(\mathfrak{R}_{\mathrm{up}})$.

Table I: Comparison between the 30PM representation of confluent Heun solutions and MST/Numerical results for $\omega = 0.1, \ell = 2$.

| $\mathfrak{R}_{\text{in}}$ | $r = 5$ | $r = 100$ |
|---|---|---|
| HYP | -0.004086908784 -i 0.000826769654 | 252.949132499007 -i 211.392731187846 |
| MST/NUM | -0.004086908784 -i 0.000826769654 | 252.949132499009 -i 211.392731187848 |
| $\mathfrak{R}_{\text{up}}$ | $r = 5$ | $r = 100$ |
| HYP | 2632.048432 -i 840.387207 | -15205.329855167 -i 989985.766975648 |
| MST/NUM | 2632.048432 -i 840.387207 | -15205.329855168 -i 989985.766975651 |

Table II: Convergence of the luminosity results. We display the dependence on the PM order $k$ of the ratio $\eta_{2,2}/\eta_{2,2}^0$ computed using the hypergeometric representation. The results are compared against MST/ NUM results displayed in the last line.

| $k$ | $v = 0.1$ | $v = 0.3$ | $v = 0.4$ |
|---|---|---|---|
| 5 | **0.961483706**187661 | **0.841**473688330721 | **0.88**7926055076 |
| 510 | **0.961483705502**195 | **0.841176**615502628 | **0.880**582026053 |
| 15 | **0.961483705502**172 | **0.841176**157785812 | **0.880**686334361 |
| 20 | **0.961483705502**172 | **0.84117617**311697 | **0.880705**358744 |
| 25 | **0.961483705502**172 | **0.84117617132**5155 | **0.880705412**455 |
| 30 | **0.961483705502**172 | **0.84117617132506**5 | **0.880705411**714 |
| MST/NUM | **0.961483705502**277 | **0.841176171325**069 | **0.880705411627** |

- **PNexpansions.nb** computes the PN expansions of $R_+(y)$, $\gamma$, $\gamma_\kappa$ at any PM order lower than 10. This code imports the data files PNRpp10.m, PNgg10.m, PNgk10.m and PNaa10.m, containing the expansions of the quantities $R_+(y)$, $\gamma$, $\gamma_\kappa$ and $a$ at order 10PM.

## V. THE HEUN EQUATION

The generalization of the previous results to the case of the non-confluent Heun equation is straightforward. The Heun differential equation is defined as

$$G''(z) + \left( \frac{\gamma_0}{z} + \frac{\gamma_1}{z-1} + \frac{\gamma_2}{z-x} \right) G'(z) + \frac{-\sigma + z\beta_0\beta_1}{z(z-1)(z-x)} G(z) = 0 \tag{80}$$

The parameters satisfy the linear relation $\gamma_0 + \gamma_1 + \gamma_2 = 1 + \beta_0 + \beta_1$. The characteristic exponents around $z \approx x$ are $(0, 1-\gamma_2)$ and at infinity are $(\beta_0, \beta_1)$. The solutions of the Heun equation can be written as linear combinations of

$$\begin{aligned} \text{Heun}_-(z) &= \text{Heun}\left[x, \sigma, \beta_0, \beta_1, \gamma_0, \gamma_1, z\right] \\ \text{Heun}_+(z) &= z^{1-\gamma_0} \text{Heun}\left[x, \sigma + (1-\gamma_0)(\gamma_2 + \gamma_1 x), 1+\beta_0-\gamma_0, 1+\beta_1-\gamma_0, 2-\gamma_0, \gamma_1, z\right] \end{aligned} \tag{81}$$

where $\text{Heun}\left[x, \sigma, \beta_0, \beta_1, \gamma_0, \gamma_1, z\right]$ is the Heun function which is normalized to one at $z = 0$ so that

$$\text{Heun}_-(z) = 1 + \dots, \ \text{Heun}_+(z) = z^{1-\gamma_0}(1 + \dots) \tag{82}$$

The dots stand for corrections in $z$. The expansion coefficients of $\text{Heun}_\alpha(z)$ can be obtained recursively, solving the equation order by order in $z$. As before, we find it convenient to parameterize the characteristic exponents

Table III: Luminosity $\eta_{2,2}/\eta_{2,2}^0$. Comparison between the different methods.

| Method | $v = 0.1$ | $v = 0.3$ | $v = 0.4$ |
|--------|-----------|-----------|-----------|
| HYP | 0.961483705502172 | 0.841176171325065 | 0.880705411714 |
| MST/NUM | 0.961483705502277 | 0.841176171325069 | 0.880705411627 |
| 22PN | 0.961483705502172 | 0.841176171325674 | 0.880705608981 |

Table IV: Total luminosity $d\mathcal{E}/dt/(\mu^2\eta_{2,2}^0)$. Comparison between the different methods.

| Method | $v = 0.1$ | $v = 0.3$ | $v = 0.4$ |
|--------|-----------|-----------|-----------|
| Hyp | 0.974719282291874 | 0.950003411484251 | 1.10991197128 |
| MST/NUM | 0.974719282291999 | 0.950003411484254 | 1.10991197120 |
| 22PN | 0.974719282291877 | 0.950003411477562 | 1.10990619336 |

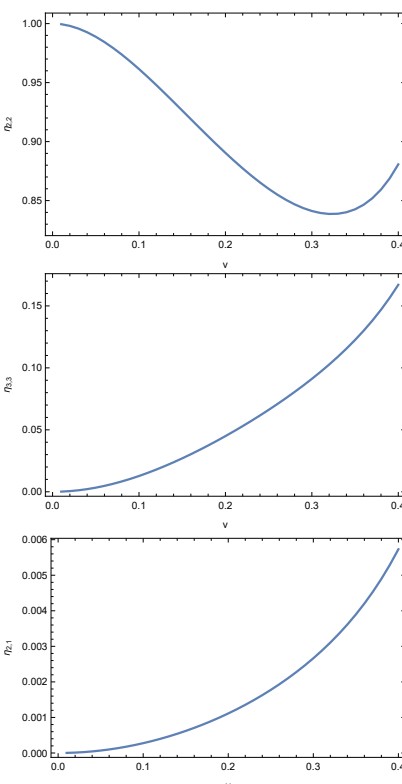

Figure 2: Luminosity vs velocity. We display the $v$-dependence of the ratio $\eta_{\ell m}/\eta_{22}^0$ for the three dominating modes. a: (2,2), b: (3,3), c: (2,1).

$\beta_i, \sigma$ of the Heun function in terms of the variables $\kappa_i, \widehat{u}$, defined as

$$
\begin{aligned}
\beta_0 &= \kappa_1+\kappa_3\,,\ \beta_1 = \kappa_1+\kappa_2\,,\ \sigma = \kappa_1^2 - \widehat{u}\,, \\
\gamma_1 &= \kappa_2+\kappa_3\,,\ \gamma_2 = 1-\gamma_0 + 2\kappa_1
\end{aligned}
\tag{83}
$$

The mathematical problem we are interested in is looking for a solution in an interval, let us say $z \in [0,1]$, satisfying

some specific boundary conditions. The general solution

$$
G(z) = \sum_\alpha B_\alpha \operatorname{Heun}_\alpha(z)
\tag{84}
$$

behaves, near the boundaries, as

$$
\begin{aligned}
G(z) &\underset{z\to 1}{\approx} D_-(1+\ldots) + D_+(1-z)^{1-\kappa_2-\kappa_3}(1+\ldots) \\
G(z) &\underset{z\to 0}{\approx} B_-(1+\ldots) + B_+z^{1-\gamma_0}(1+\ldots)
\end{aligned}
\tag{85}
$$

Specifying boundary conditions corresponds to fix two linear combinations made out of the four coefficients $B_\pm$, $D_\pm$.

## A. The hypergeometric representation

As before we start from the PM/PN ansätze

$$
G_\alpha(z) = G_\alpha^0(\tfrac{x}{z})g_\alpha(x) + \ldots = G_\alpha^1(z) + \ldots
\tag{86}
$$

with

$$
\begin{aligned}
G_\alpha^0(y) &= P_0(y)\,H_\alpha^0(y)+\widehat{P}_0(y)\,yH_\alpha^{0\prime}(y) \\
G_\alpha^1(z) &= P_1(z)\,H_\alpha^1(z)+\widehat{P}_1(z)\,zH_\alpha^{1\prime}(z) \\
H_\alpha^0(y) &= y^{\kappa_1-\alpha a}{}_2F_1(\kappa_1-\alpha a, 1+\kappa_1-\gamma_0-\alpha a, 1-2\alpha a, y) \\
H_\alpha^1(z) &= z^{\alpha a-\kappa_1}{}_2F_1(\alpha a+\kappa_2, \alpha a+\kappa_3; 1+2\alpha a; z)
\end{aligned}
\tag{87}
$$

and

$$
P_0(y) = 1+\sum_{i=1}^{k}\sum_{j=0}^{i-1}c_{ij}\,x^iy^{j-i}\,,\ \widehat{P}_0(y) = \sum_{i=1}^{k}\sum_{j=0}^{i}\widehat{c}_{ij}\,x^iy^{j-i}
\tag{88}
$$

$$
P_1(z) = 1+\sum_{j=1}^{k}\sum_{i=0}^{j-1}d_{ij}\,z^{i-j}x^j\,,\ \widehat{P}_1(z) = \sum_{j=1}^{k}\sum_{i=0}^{j}\widehat{d}_{ij}\,z^{i-j}x^j
$$

We notice that now only the non-confluent hypergeometric functions are involved. As before, plugging the

ansätze (86) into (80) and setting to zero the coefficients of $H_\alpha^p$ ($p = 0, 1$) and its derivatives, one finds a linear system of algebraic equations for the coefficients $c_{ij}$, $\widehat{c}_{ij}$, $d_{ij}$, $\widehat{d}_{ij}$. For example, for $k = 1$ one finds

$$
\begin{aligned}
c_{10} &= \frac{\left[2a^2\left(\kappa_1+\kappa_2+\kappa_3-\frac{1}{2}\right)+2\kappa_1\kappa_2\kappa_3-\kappa_1\kappa_2-\kappa_1\kappa_3-\kappa_2\kappa_3\right]}{4a^2-1} \\
\widehat{c}_{11} &= -\widehat{c}_{10} = \frac{\left(2a^2-\kappa_2-\kappa_3+2\kappa_2\kappa_3\right)}{4a^2-1} \\
d_{01} &= 1 - \frac{y(2\gamma_0-2\kappa_1-1)\left(a^2-\kappa_1^2\right)}{4a^2-1} \\
\widehat{d}_{11} &= -\widehat{d}_{01} = \frac{\left(2\kappa_1^2+2a^2-2\gamma_0\kappa_1+\gamma_0-1\right)}{4a^2-1}
\end{aligned}
$$

and, writing $\widehat{u} = a^2 + \sum_i \widehat{u}_i x^i$, we get

$$
\widehat{u}_1 = \frac{\gamma_0\left[a^2\left(1-2\kappa_1-2\kappa_2-2\kappa_3\right)+\kappa_1\kappa_2+\kappa_1\kappa_3+\kappa_2\kappa_3-2\kappa_1\kappa_2\kappa_3\right]}{4a^2-1} - \frac{\left(a^2-\kappa_1^2\right)\left(2a^2-\kappa_2+2\kappa_2\kappa_3-\kappa_3\right)}{4a^2-1}
$$

As in the confluent case, we denote by $g_\alpha$ the ratio $\frac{G_\alpha^1(z)}{G_\alpha^0(\frac{x}{z})}$. Explicit computations show that, as in the confluent case, the ratio of the two components can be written as

$$
\gamma(x) = \frac{g_+(x)}{g_-(x)} = x^{2a}e^{-\partial_a\mathcal{F}_{\text{inst}}(a,x)} \tag{89}
$$

with $\mathcal{F}_{\text{inst}}$ still is given by (18), (12) and

$$
u(a) = \frac{\widehat{u}(a) + x\left[\kappa_2\kappa_3 + (\gamma_0 - \kappa_1)(\kappa_2 + \kappa_3)\right]}{1 - x} \tag{90}
$$

The function $\mathcal{F}_{\text{inst}}$ corresponds now to the prepotential of a $\mathcal{N} = 2$ supersymmetric $SU(2)$ gauge theory with four fundamental flavours.

## B. Connection formulae

As in the confluent case, connection formulae for the Heun function can be easily derived from the standard hypergeometric transformation rules of $H_\alpha^p$. We introduce the two extra sets of hypergeometric functions

$$
\begin{aligned}
\widetilde{H}_\alpha^0(y) &= e^{i\pi\frac{1+\alpha}{2}(1-\gamma_0)}y^{\frac{1+\alpha}{2}(\gamma_0-1)}{}_2F_1\left(\tfrac{1+\alpha}{2}(1-\gamma_0)+\kappa_1+a, \tfrac{1+\alpha}{2}(1-\gamma_0)+\kappa_1-a, 1+\alpha\left(1-\gamma_0\right); y^{-1}\right) \\
\widetilde{H}_\alpha^1(z) &= z^{a-\kappa_1}(1-z)^{\frac{1+\alpha}{2}(1-\kappa_2-\kappa_3)}{}_2F_1\left(\tfrac{1+\alpha}{2}+a-\alpha\kappa_2, \tfrac{1+\alpha}{2}+a-\alpha\kappa_3, 1+\alpha\left(1-\kappa_2-\kappa_3\right); 1-z\right)
\end{aligned} \tag{91}
$$

We notice that both $\widetilde{H}_\alpha^0(y)$ and $\widetilde{H}_\alpha^1(z)$ are invariant under the sign flipping $a \to -a$, due to hypergeometric identities. The functions (91) are related to those introduced in (87) by the standard hypergeometric relations

$$
H_\alpha^0(y) = \sum_{\alpha'=\pm} B_{\alpha\alpha'}\widetilde{H}_{\alpha'}^0(y), \quad H_\alpha^1(z) = \sum_{\alpha'=\pm} F_{\alpha\alpha'}\widetilde{H}_{\alpha'}^1(z) \tag{92}
$$

with

$$
\begin{aligned}
B_{\alpha\alpha'} &= \frac{e^{i\pi(\kappa_1-\alpha a)}\Gamma(1-2\alpha a)\Gamma(\alpha'(\gamma_0-1))}{\Gamma\left(\frac{1-\alpha'}{2}-\alpha a+\alpha'(\gamma_0-\kappa_1)\right)\Gamma\left(\frac{1-\alpha'}{2}-\alpha a+\alpha'\kappa_1\right)} \\
F_{\alpha\alpha'} &= \frac{\Gamma(1+2\alpha a)\Gamma(\alpha'[\kappa_2+\kappa_3-1])}{\Gamma\left(\frac{1-\alpha'}{2}+\alpha a+\alpha'\kappa_2\right)\Gamma\left(\frac{1-\alpha'}{2}+\alpha a+\alpha'\kappa_3\right)}
\end{aligned} \tag{93}
$$

Plugging (92) into (87) one finds the connection formulae

$$
\begin{aligned}
G_\alpha^0(y) &= \sum_{\alpha'=\pm} B_{\alpha\alpha'}\widetilde{G}_{\alpha'}^0(y), \\
G_\alpha^1(z) &= \sum_{\alpha'=\pm} F_{\alpha\alpha'}\widetilde{G}_{\alpha'}^1(z)
\end{aligned} \tag{94}
$$

with

$$
\begin{aligned}
\widetilde{G}_\alpha^0(y) &= \left[P_0(y)\widetilde{H}_\alpha^0(y)+\widehat{P}_0(\tfrac{x}{z})y\widetilde{H}_\alpha^{0\prime}(y)\right] \\
\widetilde{G}_\alpha^1(z) &= \left[P_1(z)\widetilde{H}_\alpha^1(z)+\widehat{P}_1(z)z\widetilde{H}_\alpha^{1\prime}(z)\right]
\end{aligned} \tag{95}
$$

describing the solution in the *far region* and *near horizon zone*. The pairs $G_\alpha^0(z)$, $G_\alpha^1(z)$, $\widetilde{G}_\alpha^0(z)$, $\widetilde{G}_\alpha^0(z)$ provide four different bases of solutions of the Heun equation. Each basis covers a different patch of the interval $z \in [0, 1]$. They are related to each other by the connection formulae (94). Remarkably the Heun connection formulae (94)

take the familiar hypergeometric form, with the important difference that the argument of the gamma functions depends on the non-trivial function $a(u)$. Finally, we notice that the Heun function corresponds to the minus component of the far region basis

$$\text{Heun}_-(z) = \widetilde{G}^0_-(z) \underset{z\to 0}{\approx} 1 + \dots \qquad (96)$$

Indeed in this limit $P_0 \to 1$, $\widetilde{H}^0_- \to 1$ and (96) follows from (95).

## VI. SUMMARY AND CONCLUSIONS

In this paper, we computed the waveform emitted by a particle moving in a Schwarzschild geometry under the assumption of low frequencies $x = 4\mathrm{i}\omega M \ll 1$. There are two ways of taking this limit, depending on whether one keeps finite the variable $z = 2M/r$ or $y = x/z = 2\mathrm{i}\omega r$. We show that in both limits, the solution at order $x^k$ of the confluent Heun equation governing the radial motion can be written as the linear sum of a single hypergeometric (or confluent hypergeometric) function and its first derivative with coefficients given by polynomials of degree $k$, determined in a recursive fashion. The results provide an (all order in velocity) formula for the PM expansion, and an (all order in the coupling) formula for the PN series.

The parameters of the hypergeometric function and the coefficients of the two polynomials are all expressed in terms of a characteristic function $a(\ell)$, that plays the role of the SW period in a gauge theory description or the "renormalized angular momentum" in the MST formulation. The connection formulae are derived from the standard hypergeometric transformation rules. The complexity of the Heun transformation rule is hidden in the non trivial dependence $a(\ell)$ (51) on the frequency and the angular momentum $\ell$.

The PM and PN representations $G^0_\alpha(z)$ and $G^1_\alpha(z)$ cover the exterior $r \gg 2M$ and interior $\omega r \ll 1$ regions of the spacetime. Given in terms of hypergeometric functions, they can be extrapolated all the way to radial infinity or to the horizon using the standard transformation rules for the hypergeometric functions. We denote by $\widetilde{G}^p_\alpha$ the bases of solutions describing these limits. Finally, we show that $G^0_\alpha(z)$ is proportional to $G^1_\alpha(z)$ and determine the coefficient of proportionality $g_\alpha$ order by order in $x$ by matching the two functions in the overlapping near zone. The final picture is summarized in the following scheme

| Near horizon | Near zone | Far zone |
|---|---|---|
| $z \approx 1$ | $x \ll z \ll 1$ | $z \ll x \ll 1$ (97) |
| $F_{\alpha\alpha'}\widetilde{G}^1_{\alpha'}$ | $G^1_\alpha = g_\alpha G^0_\alpha$ | $g_\alpha B_{\alpha\alpha'}\widetilde{G}^0_{\alpha'}$ |

with $B_{\alpha\alpha'}$ and $F_{\alpha\alpha'}$ the hypergeometric connection matrices sending $z \to 1/z$ and $z \to z+1$ respectively. The whole non-triviality of the Heun connection formula is

codified in the dependence of these matrices on the characteristic function $a(\ell)$.

The incoming and upgoing waveforms are given by linear combinations of any of these bases, see for example (63). When compared against the PN expansions that one can find in literature [7, 20], these formulae resum the infinite tower of logarithmically divergent terms (tail and tail of tails), responsible for the slow convergence of the PN series, into exponentials. They also summarise in a compact form the dependence on $\ell$ of the waveforms, all the so called tail and tail of tail terms of the multipole expansion and have no logarithmic terms (responsible for the slow convergence of the PN series). The results for the waveform have been compared against those obtained via MST and numerical methods, finding an agreement from 9 to 13 digits in the whole range of parameters allowed for bounded orbits.

Similar results are obtained for solutions of the non-confluent Heun equation. We remark that the results of this paper apply to any physical problem governed by a Heun differential equation or a confluence of it. In particular, they can be easily adapted to the case of collisions of spinning BHs (Kerr geometry) or orbital motions in the backgrounds of topological stars, D-branes, cosmological perturbations [58]. It would also be very interesting to apply our results to the generation of templates of waveforms for GWs data analysis[5]. We hope to come back to this issue in a future publication.

### ACKNOWLEDGEMENTS

F. F. and J. F. M. were partially supported by the MIUR PRIN contract 2020KR4KN2 "String Theory as a bridge between Gauge Theories and Quantum Gravity" and the INFN project ST&FI "String Theory and Fundamental Interactions". Moreover they thank M. Bianchi, M. Billò, D. Bini, G. Bonelli, M.L. Frau, A.Lerda, A.Nagar, M.Panzeri, A. Tanzini for fruitful scientific exchanges. A. C. would like to thank the Department of Theoretical Physics at CERN for kind hospitality during the final stages of the present work. The research of R. P. and supported by the Armenian SCS grants 21AG-1C060 and 24WS-1C031. H. P. received support from the Armenian SCS grants 21AG-1C062 and 24WS-1C031.

### Appendix A: Gauge/Gravity/CFT dictionary

In this Appendix, we display the dictionaries needed to translate the variables used in this paper to describe the confluent Heun equation to those used in gravity, gauge theory and in the CFT descriptions of the Heun equation.

———

[5] We thank A.Nagar and M.Panzeri for useful discussions on this issue.

In all the cases, the confluent Heun can be written in the normal form

$$\Psi''(z) + Q(z)\Psi = 0 \tag{A1}$$

with[6]

$$Q_{\mathrm{CHE}} = -\frac{x^2}{4z^4} + \frac{x - 2x\kappa_1}{2z^3} + \frac{1 - (\kappa_2 - \kappa_3)^2}{4(z-1)z} + \frac{(2 - \kappa_2 - \kappa_3)(\kappa_2 + \kappa_3)}{4(z-1)^2 z} + \frac{u - \frac{1}{4} - \frac{x}{2}(\kappa_2 + \kappa_3)}{(z-1)z^2}$$

$$Q_{\mathrm{gauge}} = -\frac{x^2}{4z^4} + \frac{xm_3}{z^3} + \frac{1 - (m_1 - m_2)^2}{4(z-1)z} + \frac{1 - (m_1 + m_2)^2}{4(z-1)^2 z} + \frac{u - \frac{1}{4} + \frac{x}{2}(m_1 + m_2 - 1)}{(z-1)z^2}$$

$$Q_{\mathrm{CFT}} = -\frac{x^2}{4z^4} - \frac{cx}{z^3} + \frac{\frac{1}{4} - p_0^2}{(z-1)z} + \frac{\frac{1}{4} - k_0^2}{(z-1)^2 z} + \frac{u - \frac{1}{4} - x\left(\frac{1}{2} - k_0\right)}{(z-1)z^2} \tag{A2}$$

$$Q_{\mathrm{gravity}} = \frac{4\omega^2 M^2}{z^4} - \frac{8i\omega M(1 + i\omega M)}{z^3} - \frac{3}{4(z-1)z} + \frac{16i\omega M(1 - i\omega M) - 3}{4(z-1)^2 z} + \frac{\ell(\ell+1) + 4i\omega M(1 + 3i\omega M)}{(z-1)z^2}$$

Matching the five coefficients one finds the dictionaries

$$
\begin{array}{llll}
\text{Gauge:} & \kappa_1 = \frac{1}{2} - m_3 & \kappa_2 = \frac{1}{2} - m_2 & \kappa_3 \to \frac{1}{2} - m_1 \\
\text{CFT:} & \kappa_1 = \frac{1}{2} + c & \kappa_2 = \frac{1}{2} - k_0 + p_0 & \kappa_3 \to \frac{1}{2} - k_0 - p_0 \\
\text{Gravity:} & \kappa_1 = \frac{5}{2} + 2i\omega M & \kappa_2 = \frac{5}{2} - 2i\omega M & \kappa_3 = \frac{1}{2} - 2i\omega M
\end{array}
\tag{A3}
$$

with

$$x = 4i\omega M, \quad u \to (\ell + \tfrac{1}{2})^2 - 4M^2\omega^2 + 10i\omega M \tag{A4}$$

---

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

---

[6] With respect to [30, 34] we find it convenient to redefine the coupling sending $x \to -x$ in $Q_{CFT}$.

*Rev. D* **51** (1995) 1646 [gr-qc/9409054].

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
