# Peer review of "Resumming Post-Minkowskian and Post-Newtonian gravitational waveform expansions"

_SciPost Physics, doi:SciPost Phys. 19, 057 (2025)_

## Round 3 · Referee Report · Anonymous (Referee 2) · 2025-6-23

Strengths

This study derives new Post-Minkowskian (PM) and Post-Newtonian (PN) corrections for specific (\ell, m) modes of the waveform in the point-particle limit.

Weaknesses

1) Although the method is presented as general, the comparisons -particularly with the MST formalism - are largely restricted to the (2,2) mode 2) The validity of the point-particle limit should be more emphasized throughout the manuscript

Report

This manuscript investigates analytically the gravitational waveforms generated by a test particle in a Schwarzschild background. The authors introduce a novel parametrization of the solutions to the confluent Heun equation, governing the perturbations sourced by a point particle in geodesic motion, in terms of hypergeometric functions, within specific kinematic regimes. This approach yields new analytical expressions for given (l, m) modes of the waveform in both the Post-Minkowskian and Post-Newtonian expansions in the probe (point-particle) limit.

The paper presents interesting and timely results. The analytic techniques are well-motivated, and the findings are potentially valuable to both the gravitational-wave and scattering-amplitude communities. There are some issues regarding the presentation of the working assumptions and clarity that should be addressed before the paper can be considered for publication in SciPost.

Requested changes

1) Extend comparison for higher multipoles 2) Through the paper (including abstract, intro, discussion), stress that the results are valid in the extreme mass-ratio limit

Recommendation

Ask for minor revision

  • validity: high
  • significance: good
  • originality: high
  • clarity: good
  • formatting: excellent
  • grammar: excellent

Author:  Francesco Fucito  on 2025-07-25  [id 5684]

(in reply to Report 2 on 2025-06-23)

THE REPLY TO REFEREE 1 ANSWERS THE REMARKS OF REFEREE 2

---

## Round 3 · Referee Report · Anonymous (Referee 1) · 2025-6-23

Strengths

1) Novel analytic methods to solve differential equations: Introduces a new parametrization of the solution to the confluent Heun equation using hypergeometric functions in relevant kinematic regimes. 2) New analytical results for scattering waveforms: Provides new Post-Minkowskian (PM) and Post-Newtonian (PN) terms for specific $(l, m)$ modes of the probe limit waveform. 3) Scientific relevance: The problem addressed is timely and the results are of great interest to the scattering amplitude and general relativity (PN,PM) communities. 4) Technically correct: Methods are appropriate and the calculations are correct within the stated approximations.

Weaknesses

1) Limited scope: Despite claiming a general method, most comparisons (e.g., to MST) focus only on the $(2,2)$ mode; little is said about higher modes. Also, the probe limit validity is not emphasized enough throughout the text. 2) Undefined notation, definition of key variables hidden in the main text: The variables $x$, $y$, and $z$ are defined inside the introduction and only inline, making the paper harder to follow. Several variables (e.g., $T$, $R^*$, $(\theta,\phi)$) are used without prior definition in Section 4.2. 3) Minor presentation issues and minor typos: Section 5 seems disconnected from the main narrative and might be better placed in an appendix; few typos in the main text

Report

The paper ``Resumming Post-Minkowskian and Post-Newtonian gravitational waveform expansions'' explores the analytic calculation of the gravitational waveform emitted by a probe particle moving in the Schwarzschild background. The authors proposed a novel way to parametrize the solution of the (confluent) Heun's equation, describing the stress-tensor perturbation of a geodesic source on top of the black hole spacetime, in terms of hypergeometric functions for some relevant kinematic regimes. In particular, they found new analytic terms in the Post-Minkowskian and Post-Newtonian expansion for some $(l,m)$ modes of the corresponding probe limit waveform.

The paper is very interesting; the methods used in the paper are appropriate, and the results obtained are of great interest to the community. However, there are several critical points which I have to raise, both regarding the presentation of the results and the organization of the paper itself, which must be addressed before I can recommend publication in SciPost. I list them in the "requested changes" section.

Requested changes

1) The comparison of the waveform and fluxes with the MST method is almost entirely done for the $(2,2)$ mode, but the introduction, title, and conclusion promise a general method for resumming (probe) waveforms. I find it important, therefore, that the authors provide some more comments about the other $(l,m)$ modes as well. How do they compare to MST? How does the performance of the newly proposed method scale at large $l$/$m$? I feel that even a partial answer to these questions in a small paragraph would greatly improve the paper.

2) Except in the abstract, the fact that the waveform and fluxes obtained in the paper are valid only in the probe limit/geodesic motion (or 1SF order, depending on the nomenclature—see Bini/Damour 1603.09175) is not clear. It would be better to emphasize this at the beginning of Section 4: $\dots$ we compute the (probe limit) waveform $\dots$'' and in the conclusion:$\dots$ exact formulae for the (probe limit) waveform $\dots$''.

3) The definition of the most important variables $x$, $y$, and $z$, on which all the PM and PN expansions in this paper are based, is hidden inside the introduction toward the end of page 2. This makes it hard for the reader to follow the presentation, having to return to it several times. I suggest writing a proper equation (not inline) in a new conventions paragraph after the ``plan of the paper,'' where additional details about the notation are also welcome (metric signature, $m_i$, $M$, $\omega$, etc.). Please refer to this equation at the beginning of the PN and PM subsections (and maybe also elsewhere, where appropriate), so that the reader can immediately understand the regime being discussed there.

4) Perhaps it is also worth stressing that the expansion being considered in Sections 2.2–2.3 is a soft one, as $x \propto \omega M \ll 1$? I think the reader would greatly appreciate some emphasis on the physical intuition behind the expansions being considered, both in the main text and in the conclusion.

5) Many variables are not defined in Subsection 4.2. What is $T$ in the waveform? What is $R^*$? And $(\theta,\phi)$? Clearly, the bulk coordinates differ from the asymptotic Bondi ones, so I feel that at least a sentence or two is definitely needed before eq.(4.10). Please check the rest of the section as well.

6) The variable $t$ in eqs.(2.1)–(2.2) then disappears in eq.(2.8)—maybe becoming $z$—but there is no equation explaining the relation. Please fix this.

7) Section 5 is not well placed in the paper, as it has nothing to do with the waveform/fluxes and there is no connection with Section 4 (and perhaps with the rest of the paper, except for the small suggestion about the relevance for the Kerr black hole in the conclusion). I suggest either moving it to an appendix or placing it earlier in the discussion of the Heun's equation.

8) Some relevant literature on the initial works on PM tree-level scattering waveforms is missing, both from the worldline (2101.12688, 2102.08339) and amplitude (2107.10193) perspectives. You may want to consider adding these references, given that the discussion in the introduction explicitly refers to the follow-up (more recent) works.

9) Please fix the alignment of the second entry of the binomial factors in eq.(4.13).

10) Grammar typos: solutio'' $\to$solution'' after eq.(6.1), Shatashvilli'' $\to$Shatashvili'' at the beginning of Section 2.1.

Recommendation

Ask for minor revision

  • validity: high
  • significance: good
  • originality: high
  • clarity: good
  • formatting: excellent
  • grammar: excellent

Author:  Francesco Fucito  on 2025-07-25  [id 5685]

(in reply to Report 1 on 2025-06-23)

1) We displayed in tables 2,3, pages 18,19, the contributions of higher (ell, m) modes and observe that, for large $\ell$ our results converge quickly than those obtained by MST methods and based on a PN expansion. We thank the referee for drawing our attention on this limit.

2) We stressed along the paper that our techniques apply to the probe limit of the binary system. See abstract, 2 paragraph of page 2, first paragraph of Sections 4 and 5.

3) As suggested by the referee we added a "convention" section after the plan that summarises our conventions. The introduction has been revisited to avoid repetitions.

4) We stressed along the paper that x->0 corresponds to the soft limit, see for example the paragraph after 2.8.

5) The coordinates introduced in (4.10) are defined in the text before 4.10 and after.

6) The relation between t and z is given in (2.6)

7) The content of section 5 has been moved to an appendix.

8) We added in the introduction some references on the initial works on PM tree level scattering waveform.

9) Aligment in 4.13 has been fixed.

10) Grammar typos at the beginning of section 2.1 are corrected.

---

## Round 3 · Referee Report · Anonymous (Referee 3) · 2025-7-8

Report

The manuscript investigates analytical representations of gravitational waveforms generated by a test particle in Schwarzschild spacetime. The Teukolsky differential equation for spin $s$ perturbations in the Schwarzschild geometry is of the confluent Heun type, and the corresponding solutions are analysed as expansions in the Newton constant (Post-Minkowskian approximation) or in the orbital velocity (Post-Newtonian approximation). In particular, the authors provide results for the probe limit of the wave form and the energy flux produced by a circular EMRI binary system up to order 30PN. The results are tested against existing 22 PN order computations available in literature, as well as numerical methods from the Black Hole Perturbation Toolkit based on the MST approach.

The notations used throughout the manuscript are in general quite heavy, and some conventions could benefit from clearer exposition. I suggest some minor corrections to improve the readability of the work.

  • In Equation (2.1) and Equation (2.2) there is a $t$ variable, whereas in Equation (2.4) (and later) this is replaced by a $z$ variable. The notation should be unified.
  • After Equation (2.12), it should be stated more clearly what is the parameter $a$, which was not introduced before. The fact that it reduces to $\sqrt{u}$ as $x\to 0$ does not seem a definition. Something more similar to a definition is given in the paragraph between Equation (2.14) and Equation (2.15), but it should be made more explicit.
  • In Equation (2.15) the authors introduce the quantity $g_{\alpha}(x)$, which is independent of $z$. This relies on the assumption that the two solutions have the same monodromy in the irregular point $z=0$. This sentence should be made more precise, what is meant for monodromy? Why this is enough to conclude that the two solutions are linearly dependent?
  • In Equation (2.22), is the numerator correct, or should it include the derivative $\partial_{m_3}$? If it is correct, I suggest to modify the sentence after Equation (2.21).
  • What do the authors mean with “the fundamental solutions” $R_{\alpha}(y)$ in Equation (3.7)? Is it just referring to the definition in Equation (3.8)? This paragraph is not clear to me, as there are two types of solutions, labelled with G and R, is the redefinition of the solutions needed to match the behaviour in Equation (3.3)?
  • There is a typo, “solutio”, after Equation (6.1).

Moreover, there are some major points to be addressed.

  • After Equation (2.13), it is said that $a$ is a function of $u$. However, in Equation (2.14) and in the following sentence, it is said that the coefficients can be obtained as expansions in $x$. It is not clear what is the definition of these expansions: are they expansions in $a$ or in $u$? And is this parameter further expanded in $x$? Some arguments along these lines seem to appear in Section 4.1, but a more pedagogical explanation should be added. This is important to understand the results in Equations (2.20) and (2.31) too.
  • The sentence before Equation (2.21) is not clear. What is special about the results in (2.20)? How are the more general relations obtained?
  • I could not find a derivation of the results in Equations (3.12)-(3.13), neither in the ancillary files. More details should be provided.
  • The sentence after Equation (3.20) deserves further explanation: can you prove that these apparent poles at integer values of $\ell$ are not poles of the combination in Equation (3.18)?
  • In Section 5, the case of the Heun equation is discussed. This contains an additional singularity at $z=\lambda$ (or $z=x$ in the following notations). There seem to be no assumptions on the location of this additional singularity, but small $x$ expansions are still performed, suggesting $x$ to be small. A first question concerns the convergence of these expansions: in the confluent Heun case it seems that $x$ only appears as a parameter in the differential equation, whereas now it plays a different role too. Does this affect the perturbative expansions? Moreover, if $x$ is small, it seems like the connection problem between the singularities at $z=0$ and $z=1$ is quite different with respect to the hypergeometric case. For example, in Reference [30], the authors provide connection formulas for the Heun equation, and the case with the fourth singularity being small gives a more complicated result.
  • Related to the previous point, what are the convergence properties of the expansions in Equation (2.11), or Equation (2.17) as $k\to\infty$? In my understanding, these questions are relevant to claim that they provide well-defined representations of the confluent Heun functions, for which the convergence properties in $t$ (referring to the notations in Equation (2.1)) are known.

In the referee’s opinion, the manuscript shows good agreement between numerical results obtained through different methods, and provides a link between different research areas. However, the above mentioned major points need to be addressed and the main results should be written more explicitly and with additional details. In particular, at the current status, I do not see in what sense the work provides a mathematical proof of the recently discovered Heun connection formulae, as stated in the abstract and in the introduction, and the analytical framework appear to require stronger foundations and justifications to be considered for publication.

Recommendation

Ask for major revision

  • validity: -
  • significance: -
  • originality: -
  • clarity: -
  • formatting: -
  • grammar: -

Author:  Francesco Fucito  on 2025-07-25  [id 5683]

(in reply to Report 3 on 2025-07-08)

  • the equation in the variable t is what you read in standard textbooks (ex. NIST math handbook). We find convenient to work on the variable z=1/t, where the irregular point is at the origin. Since we are using an unusual form of the Heun equation we thought having the two forms could be more informative.

  • we added 2.9 to clarify what the meaning of $a$ as eigenvalue of the monodromy matrix, see also 2.13 for the relation between $a$ and $u$. Solutions are eigenvectors of the monodromy action, so any two solutions with the same eigenvalue should be proportional to each other since the eigenvector space is one dimensional.

  • we write 2.20 in terms of F_inst to make its meaning more straightforward

  • we changed the sentence after 3.7 to clarify the meaning of R_alpha . We denote by $R_\alpha$ the two solutions of the Teukolsky equation 3.1, related to G_alpha via 3.8

  • typo has been corrected

major points

  • We added a sentence after (2.13) to clarify the relation between a and u. We can think of u(a) as an expansion in x with coefficients u_i(a) depending of a, or viceversa, inverting this relation as a(u)=\sqrt{u}+sum_i a_i(u) x^i. As explained after (2.13), the coefficients u_i(a) are determined imposing the differential equation order by order in x.

  • We have reworded the sentence before (2.20)

  • (3.12) is given in the ancillary file PNexpansion.nb, in the section R_+. We added also 3.13 at the end of this ancillary file.

  • We have checked it explicitly. We do not have a general proof. we have just checked it up to the orders we have reached.

  • the position of the singularities in the non-confluent case is arbitrary so there is nothing special in the new singularity (in the confluent case two singularities merge). The connection formulas in the non-confluent case take a similar form than the one in the confluent case, after replacing B_{alpha alphap } by its non confluenti analog, and F_{rm inst} by the pre potential of a gauge theory with Nf=4 flavours, that it is a bit more complicated than that of Nf=3.

  • We cannot say anything about the k->infty limit. The convergence properties of PN expansions are not known. This is an interesting question, but it goes beyond the scope of this paper.

-The connection formulas are summarised in formula (2.38), that follows immediately from the hypergeometric relations (2.22) and (2.31). We added some explanations around 2.38.

---

## Round 5 · Author Response

this is a version changed according to the referees' suggestions

---

## Round 5 · List of Changes

referees 1 & 2

1) We displayed in tables 2,3, pages 18,19, the contributions of higher (ell, m) modes and observe that, for large ℓ

our results converge quickly than those obtained by MST methods and based on a PN expansion. We thank the referee for drawing our attention on this limit.

2) We stressed along the paper that our techniques apply to the probe limit of the binary system. See abstract, 2 paragraph of page 2, first paragraph of Sections 4 and 5.

3) As suggested by the referee we added a "convention" section after the plan that summarises our conventions. The introduction has been revisited to avoid repetitions.

4) We stressed along the paper that x->0 corresponds to the soft limit, see for example the paragraph after 2.8.

5) The coordinates introduced in (4.10) are defined in the text before 4.10 and after.

6) The relation between t and z is given in (2.6)

7) The content of section 5 has been moved to an appendix.

8) We added in the introduction some references on the initial works on PM tree level scattering waveform.

9) Aligment in 4.13 has been fixed.

10) Grammar typos at the beginning of section 2.1 are corrected.

referee 3

we added 2.9 to clarify what the meaning of a as eigenvalue of the monodromy matrix, see also 2.13 for the relation between a and u

. Solutions are eigenvectors of the monodromy action, so any two solutions with the same eigenvalue should be proportional to each other since the eigenvector space is one dimensional.

we write 2.20 in terms of F_inst to make its meaning more straightforward

we changed the sentence after 3.7 to clarify the meaning of R_alpha . We denote by Rα

the two solutions of the Teukolsky equation 3.1, related to G_alpha via 3.8

typo has been corrected

major points

We added a sentence after (2.13) to clarify the relation between a and u. We can think of u(a) as an expansion in x with coefficients u_i(a) depending of a, or viceversa, inverting this relation as a(u)=\sqrt{u}+sum_i a_i(u) x^i. As explained after (2.13), the coefficients u_i(a) are determined imposing the differential equation order by order in x.

We have reworded the sentence before (2.20)

(3.12) is given in the ancillary file PNexpansion.nb, in the section R_+. We added also 3.13 at the end of this ancillary file.

We have checked it explicitly. We do not have a general proof. we have just checked it up to the orders we have reached.

the position of the singularities in the non-confluent case is arbitrary so there is nothing special in the new singularity (in the confluent case two singularities merge). The connection formulas in the non-confluent case take a similar form than the one in the confluent case, after replacing B_{alpha alphap } by its non confluenti analog, and F_{rm inst} by the pre potential of a gauge theory with Nf=4 flavours, that it is a bit more complicated than that of Nf=3.

We cannot say anything about the k->infty limit. The convergence properties of PN expansions are not known. This is an interesting question, but it goes beyond the scope of this paper.

-The connection formulas are summarised in formula (2.38), that follows immediately from the hypergeometric relations (2.22) and (2.31). We added some explanations around 2.38.

---

## Editorial Decision

published